# Compromised Epithelial Cell Attachment after Polishing Titanium Surface and Its Restoration by UV Treatment

**DOI:** 10.3390/ma13183946

**Published:** 2020-09-07

**Authors:** Takahisa Okubo, Takayuki Ikeda, Juri Saruta, Naoki Tsukimura, Makoto Hirota, Takahiro Ogawa

**Affiliations:** 1Weintraub Center for Reconstructive Biotechnology, Division of Advanced Prosthodontics, UCLA School of Dentistry, Los Angeles, CA 90090-1668, USA; okubotakahisa@gmail.com (T.O.); saruta@kdu.ac.jp (J.S.); tsukimura.naoki@nihon-u.ac.jp (N.T.); mhirota@yokohama-cu.ac.jp (M.H.); togawa@dentistry.ucla.edu (T.O.); 2Department of Partial Denture Prosthodontics, Nihon University School of Dentistry, 1-8-13 Kanda Surugadai, Chiyoda-ku, Tokyo 101-8310, Japan; 3Department of Complete Denture Prosthodontics, Nihon University School of Dentistry, 1-8-13 Kanda Surugadai, Chiyoda-ku, Tokyo 101-8310, Japan; 4Department of Oral Science, Graduate School of Dentistry, Kanagawa Dental University, 82 Inaoka, Yokosuka, Kanagawa 238-8580, Japan; 5Department of Oral and Maxillofacial Surgery/Orthodontics, Yokohama City University Medical Center, 4-57 Urafune-cho, Yokohama, Kanagawa 232-0024, Japan

**Keywords:** attachment, human oral epithelial cell, UV treatment, implant, polished, titanium, prosthetic abutments

## Abstract

Titanium-based implant abutments and tissue bars are polished during the finalization. We hypothesized that polishing degrades the bioactivity of titanium, and, if this is the case, photofunctionalization-grade UV treatment can alleviate the adverse effect. Three groups of titanium disks were prepared; machined surface, polished surface and polished surface followed by UV treatment (polished/UV surface). Polishing was performed by the sequential use of greenstone and silicon rubber burs. UV treatment was performed using a UV device for 12 min. Hydrophobicity/hydrophilicity was examined by the contact angle of ddH_2_O. The surface morphology and chemistry of titanium were examined by scanning electron microscopy (SEM) and X-ray photoelectron spectroscopy (XPS), respectively. Human epithelium cells were seeded on titanium disks. The number of cells attached, the spreading behavior of cells and the retention on titanium surfaces were examined. The polished surfaces were smooth with only minor scratches, while the machined surfaces showed traces and metal flashes made by machine-turning. The polished surfaces showed a significantly increased percentage of surface carbon compared to machined surfaces. The carbon percentage on polished/UV surfaces was even lower than that on machined surfaces. A silicon element was detected on polished surfaces but not on polished/UV surfaces. Both machined and polished surfaces were hydrophobic, whereas polished/UV surfaces were hydrophilic. The number of attached cells after 24 h of incubation was 60% lower on polished surfaces than on machined surfaces. The number of attached cells on polished/UV surfaces was even higher than that on machined surfaces. The size and perimeter of cells, which was significantly reduced on polished surfaces, were fully restored on polished/UV surfaces. The number of cells remained adherent after mechanical detachment was reduced to half on polished surfaces compared to machined surfaces. The number of adherent cells on polished/UV surfaces was two times higher than on machined surfaces. In conclusion, polishing titanium causes chemical contamination, while smoothing its surface significantly compromised the attachment and retention of human epithelial cells. The UV treatment of polished titanium surfaces reversed these adverse effects and even outperformed the inherent bioactivity of the original titanium.

## 1. Introduction

Peri-implant mucositis and peri-implantitis are the major problems in modern dental implant therapy. Not only is the inflammation that leads to bone resorption and implant failure a serious issue, but so are the morphological changes, such as soft tissue recession and marginal bone loss, that create aesthetic issues. There are more than 1 × 10^9^ bacteria in the supragingival area of one tooth, while subgingivally, the bacteria can reach 1 × 10^3^ in a healthy shallow sulcus and 1 × 10^8^ in a periodontal pocket [1]. Protecting peri-implant tissues from the infection of these bacteria is the key to preventing peri-implant diseases [2].

The intimate contact and strong adhesion of soft tissues may provide a seal around implant abutments and prosthetic components, and prevent bacterial invasion [3]. Initial epithelial healing following non-surgical and surgical periodontal intervention appears to be achieved after 7 to 14 weeks, while the establishment of the biological width and barrier function around transmucosal implants or abutments requires 6 to 8 weeks [4]. To this end, the interaction between epithelial cells and titanium materials holds the key to the successful establishment of a peri-implant seal.

We here focus on titanium-based prosthetic abutments and tissue bars for their ability to attract epithelial cells, facilitate their settlement, and retain them once they settle. Prosthetic abutments, whether temporary or final abutments, as well as tissue bars are polished before being placed in patients [5]. Bacterial adhesion is positively correlated with surface roughness [6,7], so areas that may be exposed to saliva need to be smooth. The polishing procedure is to make the surfaces smooth and is presumed to prevent bacterial attachment. Polishing may produce chemistry changes on the base materials due to the necessary contact with polishing burs; however, what type of chemistry changes that happen, and their degree, are unclear, and more importantly, whether the chemistry change affects the biological capability of the materials is unknown.

The enhanced biological activity of UV-treated titanium has been demonstrated in osteoblasts and bone. UV-photofunctionalization is a surface treatment technique for enhancing the ability of dental implants to integrate with bone [8,9]. Implants are treated with UV light immediately prior to use. UV treatment converts titanium surfaces from hydrophobic to hydrophilic, and removes hydrocarbons that have been deposited naturally and unexceptionally [10,11]. Animal studies showed that the strength of osseointegration is increased three times during the early stage of healing by UV treatment, and the bone-and-implant contact increases from 45-55% to higher than 95% [8]. We hypothesized that if the above-mentioned polished titanium shows degraded biological capability, the ability of UV treatment to remove carbon-containing impurities may resolve or alleviate the potential adverse effect. The objective of this study was to examine the attachment, spreading behavior and retention of human epithelial cells on machined titanium surfaces, polished titanium surfaces, and polished and UV-treated titanium surfaces. We also carefully examined the surface physicochemical properties of these three surfaces as a potential factor responsible for the changes in biological capability.

## 2. Materials and Methods

### 2.1. Titanium Disk Samples and Surface Characterization

Titanium disks (20 mm in diameter and 1.5 mm thickness) were made from commercially pure grade 2 titanium (Yield Strength: 40,000 psi; Hardness: Rockwell B65; Specifications Met: ASTM B348) (McMASTER-CARR, Los Angeles, CA, USA). Three groups of titanium disks were prepared; machined surface (Machined surface), polished surface (Polished surface) and polished surface followed by UV treatment (Polished/UV). Polishing was performed by sequential use of green stone and silicon rubber burs (Shofu inc, Kyoto, Japan). The UV treatment procedure was performed using a TheraBeam SuperOsseo apparatus (Ushio, Tokyo, Japan) for 12 min. The surface roughness of the specimens was measured at three random locations using a profilometer (Surfcom 1400, Tokyo Seimitsu, Tokyo, Japan) with a stylus tip (radius, 5 μm). The surface morphologies of the studied specimens were examined via scanning electron microscopy (SEM; Nova 230 Nano SEM, FEI, Hillsboro, OR, USA). In addition, the hydrophobicity values of the samples were determined by measuring the contact angle formed by 10 µL of sterile ultra-pure water (ddH_2_O). The surface roughness of the specimens was measured at three random locations using a profilometer (Tokyo Seimitsu, Tokyo, Japan). The chemical property of the titanium surface was evaluated via X-ray photoelectron spectroscopy (XPS; Axis Ultra DLD, Kratos Analytical Ltd, Wharfside, Manchester, UK) under high-vacuum conditions (corresponding to a background pressure of 6 × 10^−7^ Pa).

### 2.2. Human Oral Epithelial Cells Culture

Human oral epithelial cells (Human Oral Epithelial Primary Cell, CELPROGEN, CA, USA) were incubated in a humidified atmosphere composed of 95% air and 5% CO_2_, at a temperature of 37 °C. After reaching a confluence of 80%, the cells were detached using 0.10% trypsin-1 mM EDTA-4Na solution and seeded onto titanium disks placed on a 12-well culture dish at a density of 3 × 10^4^ cell/cm^2^. The culture medium (Human Oral Epithelial Cell Culture Complete Growth Media with Serum and Antibiotics, CELPROGEN, CA, USA) was renewed every 3 days.

### 2.3. Cell Attachment and Proliferation

The initial attachment of cells to the titanium disks was evaluated by measuring the number of cells on their surfaces after incubation for 6 and 24 h using a water-soluble tetrazolium salts (WST-1)-based colorimetric assay (Roche Applied Science, Mannheim, Germany). The amount of produced formazan was measured at a wavelength of 420 nm using an enzyme-linked immunosorbent assay (ELISA) reader (Synergy HT, BioTek Instruments, Winooski, VT, USA).

### 2.4. Morphology and Spreading of Human Oral Epithelial Cells

The spreading and cytoskeletal arrangement of human oral epithelial cells seeded onto the titanium disk surfaces were examined via confocal laser scanning microscopy (TCS SP5, Leica, Wetzlar, Germany). After 6 and 24 h of seeding, the cells were fixed in 10% formalin solution and stained with fluorescent rhodamine phalloidin dye (actin filament, red color; Molecular Probes, Eugene, OR, USA) and vinculin (green color; Abcam, Cambridge, MA, USA). The areas, perimeters, Feret’s diameters and densities of the rhodamine and vinculin positive regions were quantified using an image analyzer software (ImageJ, NIH, Bethesda, MD, USA).

### 2.5. Cell Adhesion Assay

The adhesive strength of the human epithelial cells to the titanium surfaces was evaluated by the percentage of remaining cells after mechanical detachment. Mechanical detachment was achieved by vibrating a culture dish (amplitude, 10 mm; frequency, 30 Hz) at 37 °C for 5 min. The cells remaining on the titanium disks was evaluated by measuring the number of cells on their surfaces after incubation for 6 and 24 h using a WST-1-based colorimetric assay (Roche Applied Science). The ratio of remaining cells as a percentage was calculated. Human oral epithelial cells seeded onto the titanium disk surfaces before and after detachment were examined via confocal laser scanning microscopy (TCS SP5, Leica). The cells were fixed in 10% formalin solution and stained with fluorescent rhodamine phalloidin dye (actin filament, red color; Molecular Probes).

### 2.6. Statistical Analysis

Culture assays were performed in triplicate (*n* = 3), except the cytomorphometry performed in six cells (*n* = 6). Measurements of contact angle and XPS elemental quantification were performed on three samples (*n* = 3). Differences between experimental groups were examined by one-way ANOVA. *P* < 0.05 was considered statistically significant.

## 3. Results

### 3.1. Surface Morphology of Machined, Polished and Polished and UV-treated Titanium

The machined surfaces showed typical morphology, with parallel line traces caused by machine-turning in low- and high-magnification SEM images (Figure 1A,B). Metallic flashes in irregular shapes are also seen. There were no line traces or metallic flashes on the polished surfaces. The polished surfaces appeared smooth and flat with only faint scratches in random directions. The polished/UV surfaces showed a similar morphology to the polished surfaces in both the low- and high-magnification images. The values of the Ra and Rsk were higher on the machined surface compared to the polished surface and polished/UV surface. On the contrary, the value of Sm appeared lower on the machined surface compared to the polished surface and polished/UV surface. There was no significant difference between the polished surface and polished/UV surface in the value of surface roughness (Figure 2A,B).

### 3.2. Hydrophilic/Hydrophobic Property of Titanium Surfaces

The machined surfaces were hydrophobic, with 10 µL of ddH2O remaining in the hemisphere without spreading. The contact angle of ddH_2_O was approximately 55° (Figure 3). Polished surfaces were more hydrophobic than machined surfaces with the contact angle of ddH_2_O being higher than 70°. The area of ddH_2_O spread was smaller on polished surfaces, as seen in the top- and side-view images. There was a remarkable change in the wettability of the polished/UV surfaces compared to the other two surfaces. The droplet of 10 µL of ddH_2_O spread immediately on the polished/UV surfaces, as shown in the images, with a contact angle smaller than 5°, indicating that the surfaces were hydrophilic.

### 3.3. Surface Elements of Titanium Surfaces

The machined surfaces showed clear peaks of the three elements titanium, oxygen and carbon, as anticipated in commercially pure titanium, in the XPS spectrum (Figure 4A). Polished surfaces showed a silicon peak, as depicted in the magnified spectrum in Figure 4A, in addition to the three elements. Moreover, the carbon peak appeared to be higher on polished surfaces than on machined surfaces. Like machined surfaces, the polished/UV surfaces showed the three elements of titanium, oxygen and carbon, without silicon. The carbon peaks on the polished/UV surfaces appeared lower than those on the other two surfaces. Quantitative assessment of XPS confirmed that the atomic percentage of carbon significantly increased on the polished surfaces compared to the machined surfaces (Figure 4B). The increase was substantial, being raised from 38% to 64%. More interestingly, the increased percentage of carbon was remarkably reduced after UV treatment. The percentage of surface carbon on the polished/UV surfaces was even lower than that on the machined surfaces. Silicon was exclusively detected on the polished surfaces, with an atomic percentage of 1.2%.

### 3.4. Ability of Titanium Surfaces to Allow Cell Attachment

The number of human epithelial cells attached to polished surfaces was approximately 50% lower than that of cells attached to the machined surfaces during 6 h of incubation (Figure 5A). The reduction remained significant after 24 h incubation. The number of cells attached to polished/UV surfaces was 30% greater than on machined surfaces, and 160% greater than on polished surfaces at 6 h. Thus, the number of attached cells was greatest in the order of polished/UV surfaces, machined surfaces and polished surfaces. This trend was unchanged at 24 h. These quantitative results were confirmed by qualitative assessment by confocal microscopic images (Figure 5B).

### 3.5. Spreading Behavior of Cells on Titanium Surfaces

Epithelial cells spread on the machined surfaces, as represented by the expansion of cytoplasm, the expression of the cytoskeletal protein actin (red stain in Figure 6A), and the expression of the focal adhesion protein vinculin (green stain in Figure 6A) at 6 h post-seeding. In contrast, the cells remained circular without spreading, and with little expression of actin or vinculin, on the polished surfaces. The cells on polished/UV surfaces were spindle and much larger than those on the other two surfaces, with extensive and intense expressions of actin and vinculin. At 24 h, these trends continued; the cells on the polished/UV surfaces were larger than on the other surfaces, whereas cells on polished surfaces remained un-spread. Cytomorphometric analyses demonstrated that the area, perimeter and Feret’s diameter of the cells were greatest on the polished/UV surfaces and smallest on the polished surfaces, both at 6 and 24 h (Figure 6B). Densitometric analysis showed that the expression of actin and vinculin was also highest on polished/UV surfaces, and smallest on polished surfaces at both 6 and 24 h.

### 3.6. Ability of Titanium Surfaces to Retain Cells

After 6 h of incubation, the attached cells were subjected to vibrational detachment. Approximately 50% of the attached cells remained adherent on machined surfaces, whereas only 20% of the cells remained on the polished surfaces (Figure 7A). Polished/UV surfaces allowed 73% of the attached cells to remain adherent. The results from the cells incubated for 24 h were similar; the percentage of adherent cells after detachment was three times greater on polished/UV surfaces than on polished surfaces. Confocal microscopic images confirmed these results by showing a similar number of cells before and after detachment on polished/UV surfaces, whereas fewer cells were counted after detachment on machined surfaces and, most typically, polished surfaces (Figure 7B).

## 4. Discussion

We here demonstrated that the polishing of titanium surfaces compromises their biological capability, particularly their ability to allow the attachment and spread of human epithelial cells and to retain the once-attached cells. The degree of compromise was generally two-fold or more, compared to the machined surface without polishing. More importantly, this compromise on the polished surface was improved by UV treatment, and was equal to or better than the machined surface. The number of attached cells and the percentage of retained cells were increased by approximately three times by treating polished surfaces with UV light. This study investigated the effectiveness of UV treatment, known as photofunctionalization, in the potential elimination of carbon and other impurities caused by silicone point contact, as well as the resulting restoration of biological activity.

Figure 1 and Figure 2 show that both the polished surface and the polished/UV surface were generally smooth. This suggests that the UV light treatment does not affect the titanium’s surface structure. Figure 3 shows that both the machined surface and polished surface were hydrophobic, with an even higher degree of hydrophobicity observed on the polished surface. In contrast to these two titanium disks, the polished/UV titanium disk was superhydrophilic. Figure 4 shows that the amount of carbon on the polished surface was higher than the amount of carbon on the machined and polished/UV surfaces. The adhesion of silicon was detected on the polished surface. On the polished/UV surface, the amount of carbon decreased and silicon was not detected. Figure 5 and Figure 7 showed that the polished surface exhibited a weaker adhesion of human epithelial cells to its surface as compared to the machined surface, and that the strongest cell adhesion was observed in the polished UV surface. According to Figure 6, the size and perimeter of the human epithelial cells adhered to the polished surface were greater when compared to the machined surface; however, their greatest magnitudes were obtained by the polished UV surface. We believe that these findings are reliable and compelling to conclude on the adverse effects of the silicon polishing of titanium surfaces, and the biologically significant effects of UV treatment.

The biological effects of photofunctionalization may be construed as an effective measure against the commonly understood drawbacks of rough surfaces. However, the surface tested in the study was smoother than the machined surface. Although UV treatment is also effective on machined surfaces [8,12], the effect of UV treatment on surfaces that were smoother than machined surfaces was not clear. The results show that UV treatment is effective not only for rough surfaces but also for polished surfaces. The current results showed the enhanced expression of actin and the focal adhesion molecule vinculin, and enhanced formation of cytoplasmic processes [13]. Such enhanced early behavior is known to promote cell colonization and stimulate growth factor production, thereby promoting cell differentiation and multiple functions. An enhanced cascade of such events may independently and jointly contribute to differentiation, with increased cell–cell interactions due to increased cell density. Furthermore, the cell viability of osteoblasts did not differ between UV-treated and untreated titanium discs [12,14]. The XPS results of the polished titanium discs investigated this time were similar to those of the aged titanium discs, except for a very small amount of silica. From this result, it is considered that there is no difference in cell viability between polishing and UV treatment. As shown in this study, enhanced and facilitated cell spreading and adhesion can be expected to enhance the subsequent peri-implant seal.

The surfaces of titanium implants are chemically contaminated by the unavoidable deposition of hydrocarbons in a time-dependent manner, whether for experimental or commercial use. The titanium surface loses hydrophilicity in a time-dependent manner after surface treatment because oxygen-containing hydrocarbons accumulate. This phenomenon is called "biological aging of the titanium", and UV treatment can clear the accumulation [8,15,16]. We have previously reported that UV treatment removes the hydrocarbons accumulated on the titanium surface and reduces the carbon content on the surface [8,9,10,11,17,18,19]. Ultraviolet light breaks the bonds between the oxygen atoms of TiO_2_ and the carbon atoms of hydrocarbons. Furthermore, the UV treatment of titanium cleans these surfaces by photocatalysis via TiO_2_ [8,9,15]. In other words, carbon is reduced by both direct decomposition via UV light and photocatalysis via titanium oxide. In addition, UV light irradiation does not change the morphology of the titanium surface because no physical stress is generated [20]. The UV light treatment of titanium are very different from other surface treatments. The XPS analysis results show that silicon polishing increased carbon and silicon content, and exposure to UV light decreased carbon content on the polished surface. The silicon points used in this study consisted of silicone rubber and silicon carbide. Therefore, the elements that were present on the polished silicon surface were Ti and O, as titanium oxide, silicon and accumulated carbon. Usually, the carbon content on the titanium surface immediately after mechanical polishing or acid treatment is reduced. These processes involve physically or chemically removing one layer of the titanium surface. However, polishing with silicon increased the amount of carbon. As a result, it was found that polishing the surface of titanium with silicon has the same effect as the aging of titanium over time.

It is generally known that the main reason for implant loss is peri-implantitis, which resembles periodontal disease. Peri-implantitis is an infection caused by bacteria in the tooth biofilm [21]. The microbiota associated with peri-implant disease are mixed anaerobic, and include species such as *Peptostreptococcus* or *Staphylococcus*, similar to the subgingival microbiota of chronic periodontitis [4,5,22]. Peri-implantitis has a high prevalence and is a detrimental factor in the long-term use of implants [23,24]. However, there is currently no standard treatment. Therefore, the sealing of the soft tissue around the implant and the resulting protection is important for the long-term use of the implant [25]. This is because the seal around the implant can resist the invasion of the above bacteria. Moreover, UV-photofunctionalization improves cell behavior and function, which are impaired by saliva contamination [26], and UV-treated surfaces exhibit significantly reduced bacterial attachment and subsequent biofilm formation [27]. These results show the decontamination effect and the antibacterial effect of the UV treatment of titanium. These UV decontamination effects and antibacterial effects will also be factors that favor the long-term use of implants. The titanium-based implant abutments examined in this work are usually exposed to bars and points. In the future, we plan to study the biological effects of contact with these devices and their mitigation by UV treatment, which must be complemented with clinical and operational testing in vivo.

## 5. Conclusions

This study revealed that polishing with silicon leads to carbon accumulation and impedes epithelial cell adhesion and growth. We also found that UV treatment improved the adverse effects of silicon polishing. This knowledge is an important discovery in implant therapy, and serves as a guideline for determining the surface morphology of implants and treating peri-implantitis.

## Figures and Tables

**Figure 1 materials-13-03946-f001:**
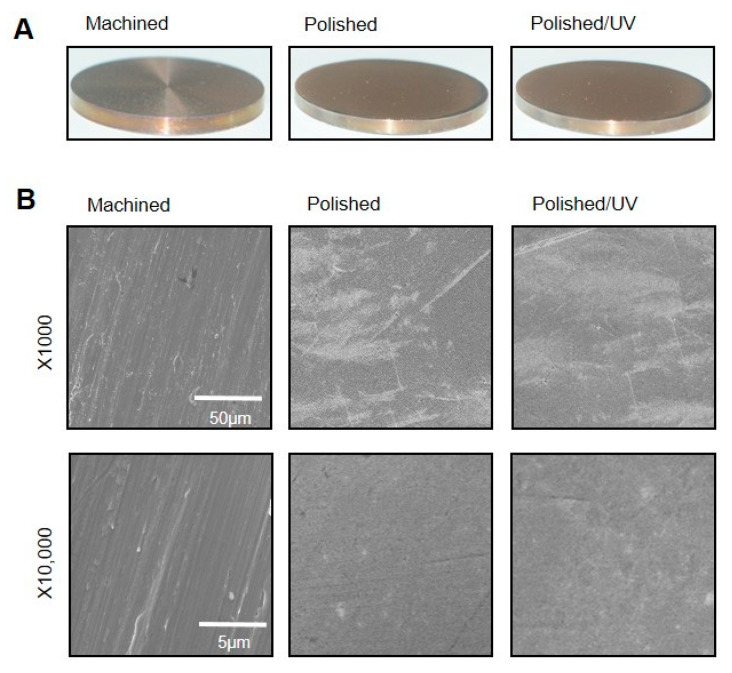
Surface morphology of the titanium disks used in this study. (**A**) Samples of the titanium disks. (**B**) Scanning electron microscopy (SEM) images of the machined surface, polished surface and polished/UV surface. Scale bar = 50 μm (1000 × magnification) and 5 μm (10,000 × magnification).

**Figure 2 materials-13-03946-f002:**
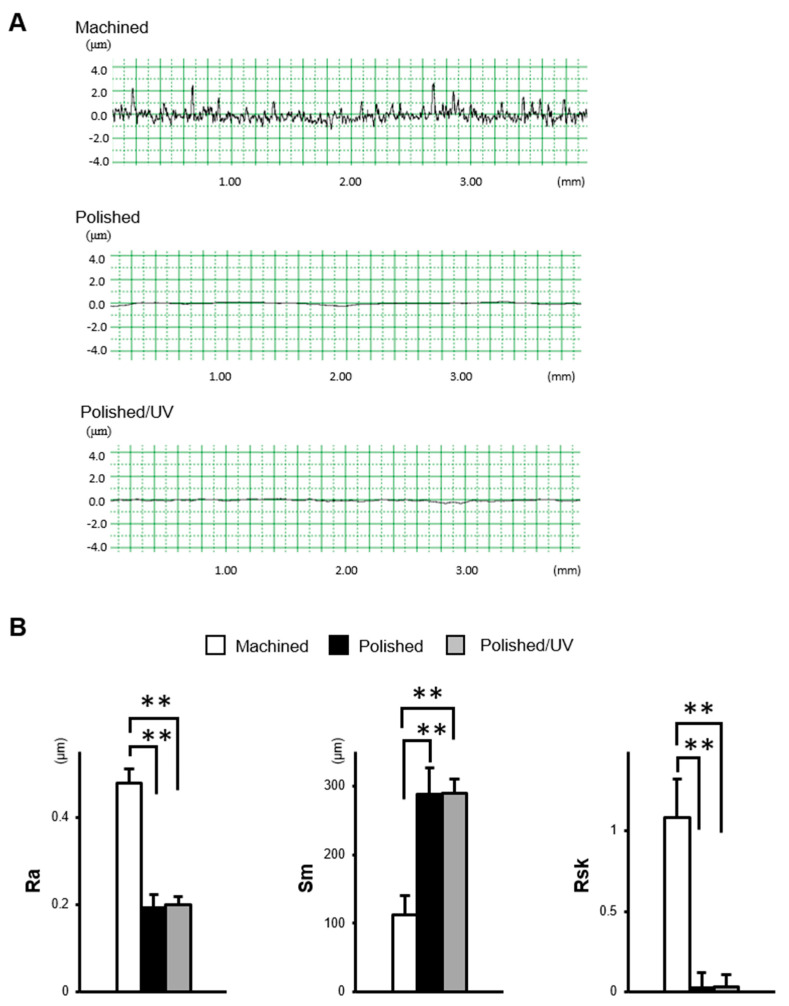
The surface characterization of titanium disks used in this study. (**A**) Two dimensional profile of titanium disks. (**B**) Quantitative topographical evaluation (histograms) of the titanium surfaces (***P* < 0.01).

**Figure 3 materials-13-03946-f003:**
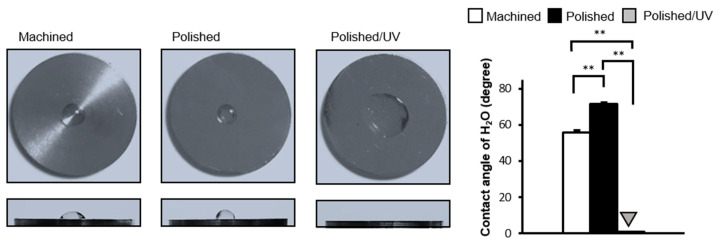
Hydrophobicity/hydrophilicity status of titanium surfaces. Contact angles for 10 µL sterile ultra-pure water (ddH_2_O) on various titanium disks (***P* < 0.01).

**Figure 4 materials-13-03946-f004:**
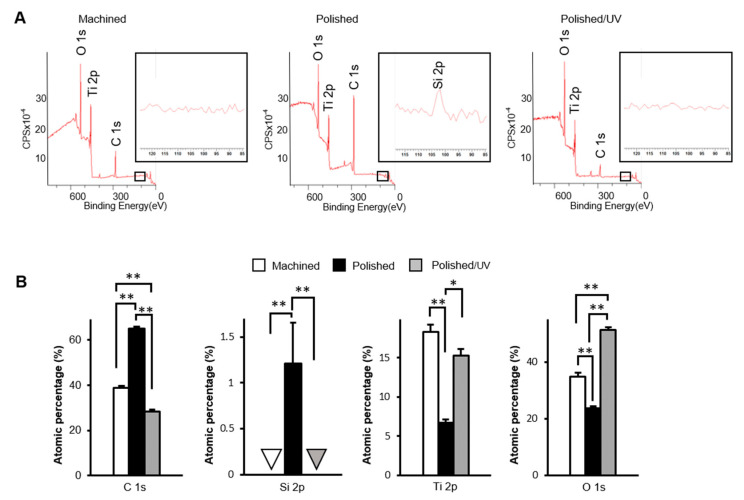
Physicochemical evaluation of titanium surfaces. The X-ray photoelectron spectroscopy (XPS) analysis of the machined surface, polished surface and polished/UV surface (**P* < 0.05, ***P* < 0.01). (**A**) Wide-scan spectrum for each samples; (**B**) Calculated atomic percentage of C1s, Si2p, Ti2p and O1s.

**Figure 5 materials-13-03946-f005:**
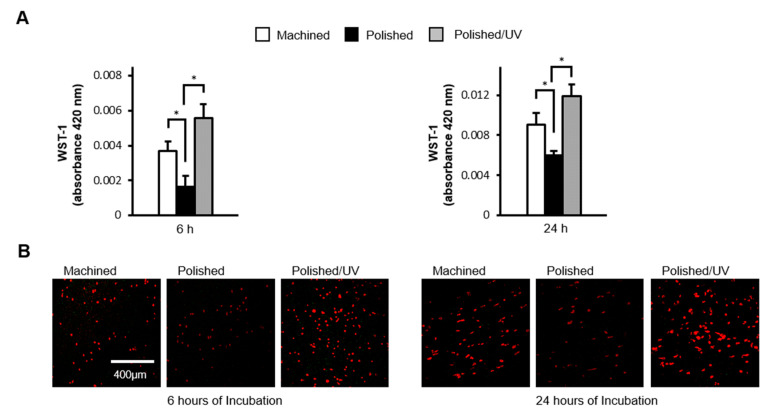
Human epithelial cell attachment to titanium surfaces during initial culture. Human epithelial cells were cultured for 6 and 24 h on the machined surface, polished surface and polished/UV surface. (**A**) Numbers of attached cells after 6 and 24 h of incubation were determined by using a WST-1 assay (**P* < 0.05). (**B**) Confocal microscopy images of human epithelial cell cultures grown on titanium disks. Representative confocal microscopy images of cells stained with rhodamine-phalloidin, for actin filaments (red). Scale bar = 400 µm.

**Figure 6 materials-13-03946-f006:**
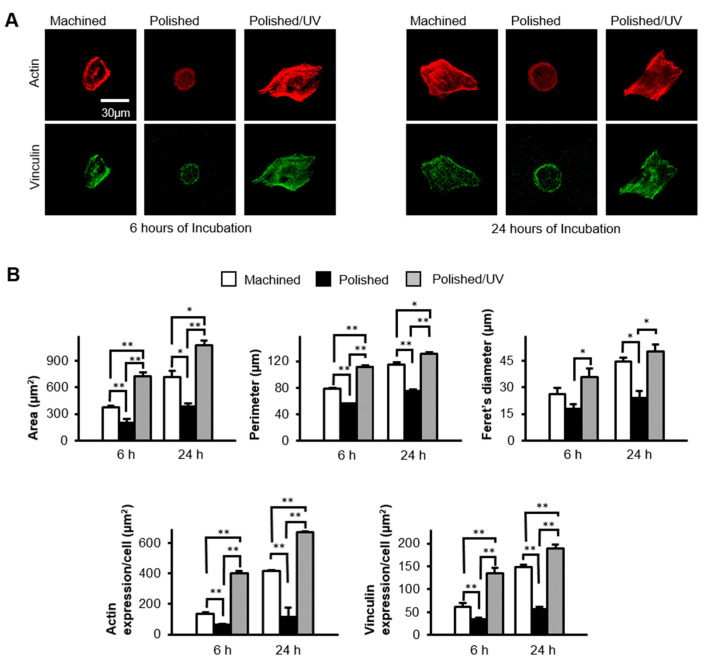
Spreading behavior of human epithelial cells on titanium disk surfaces during initial culture, after 6 and 24 h of seeding. (**A**) Confocal microscopy images of human epithelial cells stained with immunochemical cytoskeletal actin (red) and vinculin (green) dyes. Scale bar = 30 µm. (**B**) Histograms of cytomorphometric parameters, constructed from the images in panel (A) (**P* < 0.05, ***P* < 0.01).

**Figure 7 materials-13-03946-f007:**
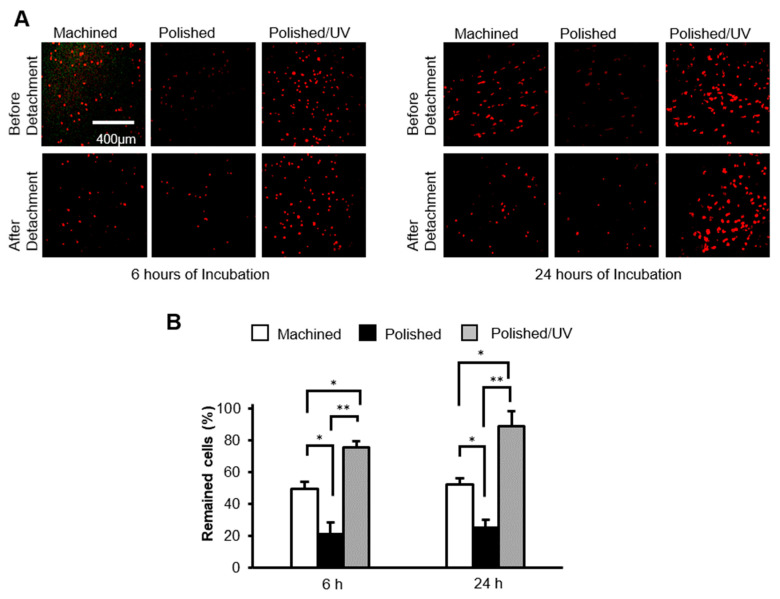
The number of human epithelial cells remaining after mechanical detachment procedures. (**A**) After 6 and 24 h of seeding, confocal microscopy images of human epithelial cell cultures grown on titanium disks. Representative confocal microscopy images of cells stained with rhodamine-phalloidin, for actin filaments (red). Scale bar = 400 µm. (**B**) The percentages of cells remaining after 6 and 24 h of incubation were determined by using a water-soluble tetrazolium salts (WST-1) assay (**P* < 0.05, ***P* < 0.01).

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
