# Peer review of "Compromised Epithelial Cell Attachment after Polishing Titanium Surface and Its Restoration by UV Treatment"

_materials, 2020, doi:10.3390/ma13183946_

Round 1

Reviewer 1 Report

Dear autors,

The manuscript deals with an interessting topic related to the osseointegration of potential biomaterials measured by the cell attachment. There many questions to be discussed, mainly because most od the authors defend rougher surface for increasing cell attachment:

  1. How were the disk prepared?
  2. Which type of Titanium was employed?
  3. What about the mechanical properties of the disk? would they be able to support mechanical efforts?
  4. The manuscript describes cells behaviour, and bacteria? The presence of bacteria is one of the main reason associated to implants fails few days after the surgery.
  5. Which advantages presented UV vs other techniques for surface modification, i.e.: DIS, femtosecond...?
  6. The studies are performed for roughness around fews microns, while bibliography discusses about 100 microns (https://doi.org/10.4028/www.scientific.net/MSF.726.39; Metals 2019, 9, 1039;).  More discussion about the suitable size of the roughness should be included.
  7. What about to treat machined surfaces with UV?

Author Response

Review1

・The manuscript deals with an interessting topic related to the osseointegration of potential biomaterials measured by the cell attachment. There many questions to be discussed, mainly because most od the authors defend rougher surface for increasing cell attachment.

We appreciate the time and effort you have dedicated to providing insightful feedback on ways to strengthen our paper.

  1. How were the disk prepared?

Thank you for your suggestion. We have rewritten [Titanium disks (20 mm in diameter and 1.5 mm thickness) were made from commercially pure grade 2 titanium (Yield Strength: 40,000 psi, Hardness: Rockwell B65, Specifications Met: ASTM B348)( McMASTER-CARR, LA, CA).] (p. 2, lines 90-92) to be more in line with your comments. Titanium discs are purchased and used from the companies listed above.

  1. Which type of Titanium was employed?

Thank you for your suggestion. We have rewritten [Titanium disks (20 mm in diameter and 1.5 mm thickness) were made from commercially pure grade 2 titanium (Yield Strength: 40,000 psi, Hardness: Rockwell B65, Specifications Met: ASTM B348)( McMASTER-CARR, LA, CA).] (p. 2, lines 90-92) to be more in line with your comments. We chose Grade 2 which is often used for implants.

  1. What about the mechanical properties of the disk? would they be able to support mechanical efforts?

Thank you for your suggestion. We have rewritten [Titanium disks (20 mm in diameter and 1.5 mm thickness) were made from commercially pure grade 2 titanium (Yield Strength: 40,000 psi, Hardness: Rockwell B65, Specifications Met: ASTM B348)( McMASTER-CARR, LA, CA).] (p. 2, lines 90-92) to be more in line with your comments. The disks have the above mechanical properties and have survived the experiment.

  1. The manuscript describes cells behaviour, and bacteria? The presence of bacteria is one of the main reasons associated to implants fails few days after the surgery.

Thank you for your suggestion. We have added [Moreover, UV-photofunctionalization improves cell behavior and function that are impaired by saliva contamination [26], and UV-treated surfaces significantly reduce bacterial attachment and subsequent biofilm formation [27]. These results show the decontamination effect and antibacterial effect of UV treatment of titanium. These UV decontamination effects and antibacterial effects will also be factors that favor the long-term use of implants.] (p. 11-12, lines 328-332) to be more in line with your comments. This study focused on cell adhesion, cell spreading, cytoskeletal organization, and adhesion strength of human oral epithelial cells. We are already studying bacteria. Therefore, we have added the above description.

  1. Which advantages presented UV vs other techniques for surface modification, i.e.: DIS, femtosecond...?

Thank you for your suggestion. We have rewritten [In other words, carbon is reduced by both direct decomposition by UV light and photocatalysis via titanium oxide. In addition, UV light irradiation does not change the morphology of the titanium surface because no physical stress is generated [20]. The relationship between these UV light and titanium is very different from other surface treatments.] (p. 11, lines 308-311) to be more in line with your comments. The advantage of UV treatment is that the effect described in this paper can be obtained without changing the surface structure of titanium.

  1. The studies are performed for roughness around fews microns, while bibliography discusses about 100 microns (https://doi.org/10.4028/www.scientific.net/MSF.726.39; Metals 2019, 9, 1039;).  More discussion about the suitable size of the roughness should be included.

Thank you for your suggestion. In our revisions, We have added [Bacterial adhesion is positively correlated with surface roughness [6,7], so areas that may be exposed to saliva need to be smooth.] (p. 2, lines 69-70). You have raised an important point. However, we believe that suitable size of the roughness would be outside the scope of our paper because this study discusses the effect of silicon polishing of titanium surface and UV treatment on human oral epithelial cells.

  1. What about to treat machined surfaces with UV?

Thank you for your suggestion. We have added [Although UV treatment is also effective on machined surfaces [8,12], the effect of UV treatment on surfaces that were smoother than machined surfaces was not clear.] (p. 11, lines 286-288) to be more in line with your comments. The effects of UV treatment on machined surfaces have been investigated. Therefore, we have added the above description.

Again, thank you for giving us the opportunity to strengthen our manuscript with your valuable comments and queries. We have worked hard to incorporate your feedback and hope that these revisions persuade you to accept our submission.

Reviewer 2 Report

In this manuscript, the authors rose a question whether polishing degrades the bioactivity of titanium-based implant, and addressed to by comparing three types of titanium materials; non-polished (machined), polished, and polished followed by UV treatment (polished/UV). They compared surface morphologies, hydrophilic/hydrophobic properties, and surface elements of these titanium. Furthermore, the authors compared bioactivities of these titanium in terms of cell attachment, spreading behaviors, and retention using human epithelial cells. The authors concluded that UV treatment improved the adverse effect of polishing.

This manuscript is well written, and author’s message is clear. This study will be contributed to this research field. I just have a few questions to be revised.

In Figure 5 and 7, the authors showed cell adhesion/retention of these titanium by WST-1 staining, but did not mentions/examine cell survival on these titanium surface. Do polishing or UV treatment affect cell adhesion only? Or do polishing or UV treatment affect cell viabilities?

Author Response

Review2

・In this manuscript, the authors rose a question whether polishing degrades the bioactivity of titanium-based implant, and addressed to by comparing three types of titanium materials; non-polished (machined), polished, and polished followed by UV treatment (polished/UV). They compared surface morphologies, hydrophilic/hydrophobic properties, and surface elements of these titanium. Furthermore, the authors compared bioactivities of these titanium in terms of cell attachment, spreading behaviors, and retention using human epithelial cells. The authors concluded that UV treatment improved the adverse effect of polishing.

This manuscript is well written, and author’s message is clear. This study will be contributed to this research field. I just have a few questions to be revised.

We appreciate the time and effort you have dedicated to providing insightful feedback on ways to strengthen our paper.

  1. In Figure 5 and 7, the authors showed cell adhesion/retention of these titanium by WST-1 staining, but did not mentions/examine cell survival on these titanium surface. Do polishing or UV treatment affect cell adhesion only? Or do polishing or UV treatment affect cell viabilities?

Thank you for providing these insights. We have added [Furthermore, Cell viability of osteoblasts did not differ between UV treated and untreated titanium discs [12,14] . The XPS results of the polished titanium discs investigated this time were similar to the aged titanium discs, except for a very small amount of silica. From this result, it is considered that there is no difference in cell viability between polishing and UV treatment.] (p. 11, lines 294-297) to be more in line with your comments. We agreed with your point and incorporated this proposal into our paper. We believe that the results are comparable to those of osteoblasts. This is because the XPS results on the titanium surface are not so different.

Again, thank you for giving us the opportunity to strengthen our manuscript with your valuable comments and queries. We have worked hard to incorporate your feedback and hope that these revisions persuade you to accept our submission.

Sincerely yours,

Takayuki Ikeda DDS, PhD
Assistant Professor
Department of Complete Denture Prosthodontics,
Nihon University School of Dentistry
1-8-13 Kanda Surugadai, Chiyoda-ku 101-8310, Tokyo, Japan

Round 2

Reviewer 1 Report

The authors have addressed most of the questions.

This manuscript is a resubmission of an earlier submission. The following is a list of the peer review reports and author responses from that submission.